# *"It doesn't require blood!"*: Perceptions around non-invasive malaria testing tools in Indonesia, Peru, and Rwanda

**Vanessa Fargnoli**[1‡], **Serafina Calarco**[1‡*], **Catherine Thomas**[2], **Caroline Thomas**[3], **Claudius Novchovick Mone Iye**[3], **Valerie A. Paz Soldan**[4,5], **Amy Celeste Morrison**[4,6], **Janvier Serumondo**[7], **Ladislas Nshimiyimana**[7], **Yvonne Delphine Nsaba Uwera**[8], **Elena Marbán-Castro**[1], **Sonjelle Shilton**[1‡], **Kevin K.A. Tetteh**[1‡]

**1** FIND, Geneva, Switzerland, **2** University of Jakarta International, Jakarta, Indonesia, **3** Yayasan Peduli Hati Bangsa, Jakarta, Indonesia, **4** Asociación Benéfica Prisma, Lima, Peru, **5** Tulane University School of Public Health and Tropical Medicine, New Orleans, Louisiana, United States of America, **6** University of California, Davis, California, United States of America, **7** Rwanda Biomedical Center, Kigali, Rwanda, **8** University of Rwanda, College of Medicine and Health Sciences, Kigali, Rwanda

‡ VF and SC are Co-first authors on this work. SS and KKAT are Co-last authors on this work.
* Serafina.calarco@finddx.org

## Abstract

Malaria remains a major global health challenge. Prompt, accurate diagnosis is crucial for effective case management. Current diagnostic approaches rely on invasive or minimally invasive sampling via venous or fingerpick blood draw, posing a risk to healthcare workers via the handling of potentially infectious body fluids. They may also be a barrier to recipients, particularly in malaria-endemic areas where there is routine testing of non-symptomatic individuals. Non-invasive tests based on saliva, exhaled volatile organic compounds (VOCs), and transdermal detection have the potential to increase case detection and linkage to care while reducing biosafety risks. Knowledge gaps exist regarding the feasibility and acceptability of these tools. This qualitative study, conducted in Indonesia, Peru, and Rwanda, aimed to generate evidence from end-users around the potential adoption of non-invasive diagnostic technologies, to determine whether these technologies are fit for purpose. Between October and November 2023, 24 semi-structured interviews were conducted with stakeholders and professionals working at borders, and 16 focus group discussions were conducted with caregivers of children under 5 years, pregnant women, healthcare workers, teachers and other community members (139 participants in total). The comfort non-invasive approaches offer to recipients of care was widely noted, especially compared with blood draw, which is considered painful for children. Ease of use and rapid diagnostic capabilities enabling real-time disease diagnosis were perceived as particularly beneficial in remote areas with limited healthcare infrastructure. Device portability was seen as a game-changer. Major concerns were the lack of information regarding the accuracy of non-invasive tools compared with established methods, the

**Data availability statement:** Our protocol, as approved by the Ethics Committees of Atma Jaya, Universitas Katolik Indonesia (number: 008H/III/PPPE.PM.10.05/09/2023), Comité Institucional de Ética en Investigación de la Asociación Benéfica Prisma, Peru (number: CE0561.23), and the Rwanda National Ethics Committee (number: 244/RNEC/2023), did not include provisions for the public sharing of the full database generated during the study; neither did the participants' informed consent. As participants did not agree to us sharing their data publicly and to ensure compliance with national ethical guidelines of the countries and participant consent agreements, we did not include the dataset in a public repository. The data supporting the findings of this study are qualitative in nature and derived from in-depth personal interviews. Due to the potentially identifying content of full transcripts, sharing these data publicly would compromise participant anonymity and confidentiality. Ethical restrictions on data sharing have been imposed by the committees that approved this study. In line with these restrictions, the full interview transcripts cannot be made openly available. Researchers who wish to request access to de-identified excerpts relevant to the study findings may contact the corresponding author (serafina.calarco@finddx.org) or the Foundation for innovative New Diagnostics (FIND) (info@finddx.org, +4122 710 05 90, https://www.find-dx.org/about-us/locations/geneva-switzerland/).

**Funding:** This study has been funded by UK aid from the UK government (FCDO project number 300341-102), as well as the Swiss Development Cooperation (contract number 81071426).

**Competing interests:** The authors have declared that no competing interests exist.

lack of trust in tests targeting specimens other than blood products, and their inability to differentiate between malaria species. Early engagement with stakeholders in the development of novel interventions is essential to ensure that end products meet the needs of communities.

## Introduction

Malaria remains a major global health issue, with an estimated 249 million cases across 85 countries and causing 608,000 deaths in 2022 alone [1]. Rapid diagnosis of malaria is crucial to facilitating prompt care, reducing the risk of complications, and protecting patient health. The most common methods of diagnosing malaria involve the use of microscopy to detect whole parasites in peripheral blood or lateral flow assay (LFA)-based rapid diagnostic tests (RDTs) that detect specific parasite proteins [2]. While these methods have proven effective [3], they can be invasive and uncomfortable for recipients due to blood collection via finger prick or venous blood draw [4]. Microscopy remains the gold standard for laboratory diagnosis but requires experienced microscopists to ensure accuracy. Although automated and digital microscopes are being developed, these are not yet widely accessible [5]. In addition, there are limitations to current LFA-based RDTs. Some strains of *Plasmodium falciparum*, the deadliest malaria parasite species, have deletions in the gene that produces histidine-rich protein 2 (HRP2), the protein detected by commonly used RDTs, leading to false-negative results [6,7]. Newer technologies are being developed to overcome these challenges and improve malaria diagnosis. This includes RDTs based on the combined detection of parasite lactate dehydrogenase (pLDH) and HRP2.

The advent of non-invasive diagnostic technologies holds promise for transforming malaria detection. Non-invasive testing in the context of malaria refers to diagnostic tests that do not require the collection of blood specimens by finger prick or venous puncture, including saliva-based tests, urine-based tests, breath/skin odor detectors, and transdermal devices [8–11]. They seek to provide a painless, safe, and efficient alternative to conventional diagnostic methods and/or they can provide a rapid detection tool that can be followed up with other diagnostic methods, potentially improving care recipients' compliance and expanding the accessibility of malaria diagnostic services.

An overview of malaria diagnostics published by UNITAID [12] and a technology landscape assessment exploring non-invasive diagnostics conducted by FIND (the Foundation for Innovative New Diagnostics) in 2023 identified several non-invasive tools at different technology readiness levels (TRLs). Among the identified technologies, some are at an advanced stage of development. These include transdermal devices based on the detection of hemozoin and some endogenous biomarkers [13–15], saliva gametocyte-detecting rapid tests [16], and volatile organic compound (VOC)-based diagnostics [17].

These advanced non-invasive technologies are part of a broader trend toward a preference for rapid and accurate diagnostic tools that can provide real-time results directly at or near a care recipient's location. Known as point-of-care (POC)

technologies, such tools can reduce the turnaround time for test results compared with traditional, laboratory-based testing, enhancing outcomes for care recipients [18]. True-POC devices are self-contained and require minimal infrastructure, making them ideal for remote settings, while near-POC technologies still rely on some centralized infrastructure but offer rapid results close to a care recipient's location.

Here, we explore the perceptions of end-users regarding the use of non-invasive true-POC malaria diagnostic tests in three malaria-endemic countries: Indonesia, Peru, and Rwanda.

Malaria presents a serious public health challenge in these countries, each facing unique difficulties and employing tailored malaria control strategies. In Indonesia, malaria is most prevalent in eastern provinces such as Papua, where *P. falciparum* and *P. vivax* are the dominant malaria species, with *P. falciparum* causing most of the severe infections and *P. vivax* contributing to recurring cases [19]. Although *P. malariae* is much less common than these species, it is sporadically detected in some regions. In these areas, control efforts focus on early diagnosis, treatment, and vector control, yet limited healthcare access and infrastructure persist [20]. In Rwanda, where *P. falciparum* is the prevalent species, malaria cases and deaths have seen major reductions through the use of RDTs and community-based management, although some districts still experience high incidence rates [21]. In Peru, most malaria cases occur in the Loreto region, where *P. vivax* is the most prevalent species and *P. falciparum*, although less common, still poses a serious risk of severe illness [22]. Here, initiatives such as the Project for Malaria Control in Andean Border Areas (PAMAFRO) and the Amazonian Malaria Initiative emphasize combined treatments, insecticide-treated nets, and active case detection [23–25].

The published literature regarding non-invasive diagnostic tests for malaria is limited. A quantitative, online survey was conducted in 2017 among national representatives of malaria control programs, which focused on the perceived value and acceptance of non-invasive malaria tests using saliva and urine specimens, and a mixed methods study was conducted in 2021 among teachers, community health workers, nurses, and laboratory workers in the context of piloting a saliva-based malaria RDT [26,27]. Understanding what opinions end-users (care recipients, healthcare providers, and community members) have regarding these non-invasive diagnostic tools is crucial. It not only affects how these technologies might initially be accepted but also their long-term use in routine healthcare.

This study was designed to capture the views of a broad representation of populations affected by malaria in each of the three countries. The participants were all adults, but they were also asked for their perspectives regarding how non-invasive diagnostic tests would be acceptable to children of different ages. Engaging in discussions with key target populations, including pregnant women and caregivers of young children, is important because they are at higher risk of severe malaria, which can greatly affect the health of both mothers and children [28]. Pregnant women are more likely to experience serious complications, such as anemia and negative birth outcomes, while children under five are more prone to severe illness and death because their immune systems are still developing [29]. These groups are also the main targets of mass malaria elimination activities. By prioritizing the voices of various end-users, including key target populations, we can foster the development of diagnostic tools that are not only innovative but also truly responsive to the needs of those at the forefront of the fight against malaria.

## Materials and methods

### Ethics statement

The study protocol was approved by three ethics committees, one in each of the countries where the study was performed: Atma Jaya, Universitas Katolik Indonesia (number: 008H/III/PPPE.PM.10.05/09/2023), Comité Institucional de Ética en Investigación de la Asociación Benéfica Prisma (number: CE0561.23), Peru, and the Rwanda National Ethics Committee (number: 244/RNEC/2023). The recruitment period took place from 12-10-2023–18-11-2023 in Indonesia, from 10-11-2023–29-11-2023 in Peru, and from 30-10-2023–08-11-2023 in Rwanda. Written, informed consent was obtained from all participants before data collection. All data were anonymized.

## Study design and sites

This study combined focus group discussions (FGDs) and semi-structured interviews (SSIs) to capture the opinions and experiences of end-users, seeking to shed light on the acceptability and potential facilitators and obstacles associated with the implementation of new, non-invasive malaria detection devices. The study employed a phenomenological approach to explore participants' lived experiences and perceptions in depth. SSIs were used to obtain in-depth, individual perspectives from stakeholders and professionals working at borders, with limited availability, while FGDs were employed to capture shared experiences and group dynamics among specific groups and community members.

SSIs were conducted with stakeholders such as health professionals working on malaria policy or control programs in malaria-endemic sites and professionals working at borders who, because of human mobility, were familiar with malaria policies. The aim was to identify how the new devices could be used and optimized in different settings. FGDs were conducted with pregnant women, teachers, caregivers of children under five, healthcare workers (HCWs) and other community members. The group of "other community members" included other members of these communities who did not fall into the above categories, but who were interested in sharing their opinions on this subject. This was decided with partners in country and could include school staff, travellers, established groups such as spiritual groups, groups at healthcare centers etc. The aim was to stimulate the sharing of experiences with malaria diagnostics and identify participants' preferences and common concerns regarding the new devices, including their potential integration and use in their respective countries. The study was conducted by local researchers in each country to mitigate power and cultural differences. Participants from different types of locations were included (Table 1).

## Study populations

Community members, pregnant women, teachers, caregivers of children under five, HCWs, health professionals working at or responsible for borders (sites identified as key transmission areas), and national stakeholders and health authorities participated in the study. Pregnant women and caregivers were approached because of their increased risk of malaria. HCWs (e.g., nurses) were targeted because of their capacity to recommend malaria testing to care recipients. Teachers and community members were chosen because of their roles in the community and involvement in decision-making regarding the utility of testing in specific settings. Professionals working at borders were recruited because of their high risk of contracting malaria (from individuals crossing the border from neighboring countries or remote areas with little or no malaria control) and because they represent a potential innovative target for service delivery of these non-invasive malaria tests at borders. Finally, stakeholders were approached because of their capacity to dedicate resources and influence national policies. Eligibility criteria to participate in the study were that informants were adults (aged 18 years or more), provided informed consent, and spoke one of the languages spoken by the facilitators.

**Table 1. Locations from which study participants were enrolled, highlighting type or area (rural, urban) and endemicity (endemic, non-endemic).**

| | | Endemic | Non-endemic |
|---|---|---|---|
| Indonesia | Rural | West Papua and East Nusa Tenggara Papua | NA |
| | Urban | NA | Jakarta |
| Peru | Rural | Zungarococha (in Loreto) | NA |
| | Urban | Iquitos (in Loreto) | Tumbes Lima |
| Rwanda | Rural | Bugesera and Gasabo District (mix of rural/urban) | Kirehe and Burera District |
| | Urban | Bugesera and Gasabo District (mix of rural/urban) | NA |

NA: Not applicable: Participants were not included from these areas.

## Sampling procedures

Individuals were recruited through purposive sampling, a non-probabilistic method commonly used in qualitative research, where participants are deliberately selected based on specific characteristics or criteria relevant to the research objective [30]. The local study teams sought potential informants using a variety of means, including through networks of medical experts and schools. Efforts were made to ensure variety in the samples in terms of gender identity, socioeconomic status, and urban and rural sites. The interviewers contacted informants by phone, email, WhatsApp, or in-person visits and provided them with information about the study aims and procedures. The sample size per country included SSIs with 6–10 stakeholders and professionals working at borders, and 4–5 FGDs with 4–12 participants each (Table 2).

## Malaria testing tools

Three types of devices were assessed as hypothetical non-invasive diagnostic tools: devices based on the detection of VOCs, spectrophotometers that detect malaria through the skin, and saliva-based RDTs (Fig 1), all three types are portable. Two of the types of tests are already available as commercialized products for multiple applications, including air monitoring (e.g., for VOCs) and food quality (e.g., spectrophotometry). However, none of these devices were on the market during data collection.

The first category of devices is based on the detection of VOCs. VOCs are a heterogeneous group of low molecular weight organic chemicals. VOCs can be emitted from the human body as gases, resulting in their presence in exhaled breath, sweat, urine, and other body fluids. VOCs are produced during various metabolic processes and can vary according to an individual's health status, diet, lifestyle, and genetic factors. There is evidence to suggest that specific VOCs, or VOC profiles, may be unique to certain diseases, because the presence of pathogens can change both the level and composition of VOCs produced during infections, potentially resulting in a diagnostic signature that can be used to identify individuals with an infection [7].

The second category of devices is based on the detection, through the skin, of parasite-associated biomarkers. The devices use spectrophotometric approaches to measure the absorption or transmission of light through the skin (e.g., of the ear, arm, or palm of the hand). These are portable, handheld devices that use different regions of the light spectrum (e.g., near-infrared (NIR) or ultraviolet-visible (UV-vis)) to detect biomarkers associated with the presence of malaria parasites in the blood. The devices work by shining a beam of harmless NIR or UV-vis light through an individual's skin and measuring the amount of light transmitted or absorbed. The spectral signature generated reflects the chemical

**Table 2. Sampling by community groups per country and per qualitative method.**

| Participants | | Method | Recruitment | Malaria endemicity area |
|---|---|---|---|---|
| Stakeholders | Including high level working on travel policy borders | 3-5 SSI | Country specific Recruited by email for an online or in-person interview | Endemic and non-endemic |
| Professionals at borders | Heads and Officers of Borders and Customs | 3-5 SSI | Country specific Recruited by email (after approval by Border authorities) | Endemic and non-endemic |
| Pregnant women | Women | 1 FGD | Antenatal care facilities, in-person | Endemic |
| Caregivers of children under-five | Gender mixed | 1 FGD | Pediatric facilities and schools, in-person | Endemic |
| Teachers | Gender mixed | 1 FGD | Schools, in-person | Endemic |
| HCWs | Gender mixed | 1 FGD | Primary healthcare centers | Endemic |
| Other community members | Gender mixed | 1 - 2 FGD | Schools, established groups at spiritual/healthcare centers, travelers etc. | Endemic and non-endemic |

FGD: Focus group discussion; SSI: Semi-structured interview.

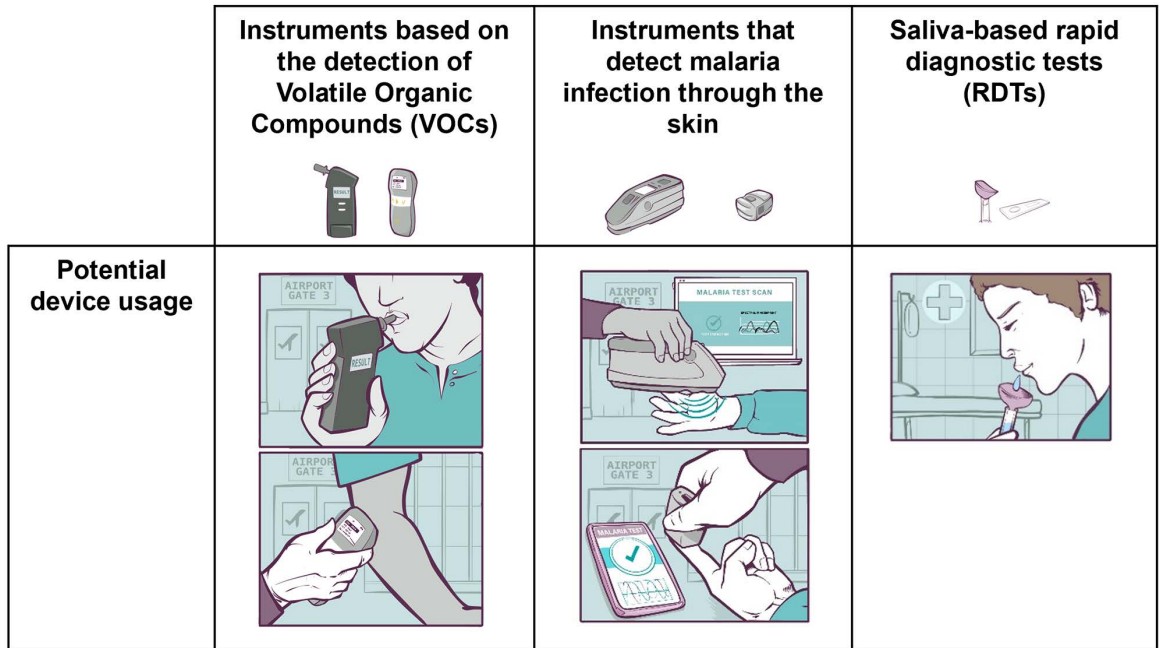

**Fig 1. Types of non-invasive devices assessed.** Visual representations of three types of hypothetical non-invasive diagnostic tools—VOCs detection devices, spectrophotometers for detecting malaria through the skin, and saliva-based RDTs—showing their potential designs and usage in practice.

composition of the blood and can be analyzed using machine learning algorithms to identify the presence of biomarkers associated with the presence of malaria parasites.

The third type of device relies on detecting the presence of malaria parasites using saliva specimens. For this type of test, the intention is that saliva is tested for the presence of gametocytes in the microvasculature, using antibodies with a high affinity to a parasite-specific protein abundant in the saliva of individuals with malaria.

### Data collection

In all three countries, the FGDs were conducted in person at clinics, schools, non-governmental organizations (NGOs), research centers, and churches. Regarding the SSIs, all of the SSIs in Indonesia were conducted online. In Peru, half of the SSIs were conducted in person and the other half online. In Rwanda, eight SSIs were conducted in person and two were held online. Only the participants and researchers were present during the FGDs and SSIs. Participants participated only in 1 SSI or FGD.

All FGDs and SSIs were conducted in vernacular languages: in Spanish in Peru by a woman senior qualitative researcher (V.P.S., PhD, social scientist and public health professor), a woman senior epidemiologist and vector-borne disease control expert (A.C.M., PhD, research scientist), and co-facilitated by a man qualitative research assistant; in Bahasa Indonesia in Indonesia by two women senior qualitative researchers (C.T.1, PhD, social scientist and C.T.2, public health expert, founder of a community-based NGO/researcher) and one man researcher (C.M., community representative, co-founder of a community-based NGO/researcher); in Kinyarwanda and English in Rwanda by four qualitative research-ers: three women (Y.D.N.U., masters in nursing (midwifery and women's health), assistant lecturer at the University of Rwanda); (J.U., bachelor of science, country director of Heart and Sole Africa/Action); (M.F.M., masters in public health, independent researcher) and two men (J.S., M.D., masters in public health) director of sexually transmitted infections (STIs); L.N., masters in public health (Epidemiology Malarial Management Senior Officer); All countries received support

from FIND: a woman senior scientist and qualitative research (V.F.) and a woman scientific lead and technology adviser on malaria (S.C.) [31]. There was no prior relationship between interviewers and participants and participants did not have previous information about the research.

The SSI and FGD guides were piloted and reviewed by the study teams to ensure cultural sensitivity and appropriate wording. The guides covered three main themes: 1) acceptability regarding the use of the new tools (in which circumstances, cultural sensitivities, and perceived benefits and disadvantages of the tools); 2) implementation of the tools (where, with which specific groups, and at what times); and 3) information about and disclosure of malaria status (including the potential stigma of others knowing an individual's malaria status). As this was an exploratory study, and prototypes were not available during the data collection period, FIND developed visual information sheets to help participants conceptualize the use of the new devices in their contexts (S1 Fig, supporting information).

All SSIs and FGDs were audio-recorded and transcribed verbatim. Notes were also taken. SSIs took between 30 minutes and 1 hour 30 minutes, and FGDs took between 45 minutes and 2 hours 15 minutes. Transcripts were cross-checked against the recordings for accuracy and completeness and subsequently translated into English. All translated transcripts in English were cross-checked with the local transcripts. The audio-recordings were destroyed once the English transcriptions had been completed. Transcripts were not returned to participants for comments or corrections. Participants did not provide feedback on the findings. Participants' sociodemographic information was collected at the end of the SSIs and FGDs.

### Data analysis

Transcripts were systematically coded using Quirkos CAQDAS, v2.5.3. The data were coded by two women (C.T.1, C.T.2) and one man researcher (C.M.) in Indonesia, three women researchers in Peru (V.P.S., E.O., L.N.), and three women (Y.D.N.U., J.U., M.F.M.) and one man (L.N.) in Rwanda. Themes were both predetermined and emergent, identified in advance, and refined through analysis of the data. Research teams at each site indicated that data saturation had been reached through the FGDs and the SSIs conducted up to November 2023, meaning no new insights were expected from further data collection. Triangulation was performed based on the responses from various end-users to create an overall narrative. The findings will be disseminated among the participants after publication and upon request. Data were reported following the COREQ (Consolidated Criteria for Reporting Qualitative Research) guidelines [31].

## Results

### Participants' characteristics

Between October and November 2023, 24 SSIs and 16 FGDs were conducted, with a total of 139 participants, aged between 18 and 64 years. Combining the demographic data of SSI and FGD participants, most participants were women (n = 95) and had received university or higher education (n = 96). Of the 139 participants, 106 were married or in a relationship, and 91 participants were working full-time (Table 3). There were 7 FGDs in Peru, 5 in Rwanda and 4 in Indonesia (Table 3).

The number of participants per FGD varied by country, with between 5 and 6 in Indonesia, between 7 and 12 in Peru, and 6 in Rwanda. The differences in the total number of participants in SSIs and FGDs among countries were due to budget and time constraints. Our findings are organized around themes for and against the new devices, followed by recommendations for their implementation, according to data collected during FGDs and SSIs for the three countries combined, except when there was a disparity between countries or groups or where a specific point was raised. The extent of their views reflects the opinions that interviewees shared spontaneously, with the interviewer remaining neutral and unbiased throughout. The Peru-specific findings are discussed in greater detail in a separate manuscript, submitted elsewhere [32].

**What does "malaria" mean to participants?.** All discussions began by asking participants for their spontaneous reactions to what malaria meant to them. *"Serious disease,"* *"deadly disease,"* *"manageable disease,"* *"seasonal illness,"* and *"mosquito bites"* were common answers across the three countries. Some participants referred to the symptoms

**Table 3. Sociodemographic characteristics of the participants.**

| | SSI (n=24) | | | | | | FGD (n=115) | | | | | | Totals | |
|---|---|---|---|---|---|---|---|---|---|---|---|---|---|---|
| | Peru | | Rwanda | | Indonesia | | Peru | | Rwanda | | Indonesia | | | |
| Number of participants | N=8 | | N=10 | | N=6 | | N=63 | | N=30 | | N=22 | | 139 | |
| Type of informant | n | % | n | % | n | % | n | % | n | % | n | % | n | % |
| Caregivers of children aged <5 years | NA | | | | | | 12 | 19 | 6 | 20 | 6 | 27 | 24 | 17 |
| Pregnant women | | | | | | | 9 | 14 | 6 | 20 | 5 | 23 | 20 | 14 |
| Healthcare workers | | | | | | | 10 | 16 | 6 | 2 | 6 | 27 | 22 | 16 |
| Other community members | | | | | | | 17 | 27 | 6 | 20 | 0 | 0 | 23 | 17 |
| Teachers | | | | | | | 15 | 24 | 6 | 20 | 5 | 23 | 26 | 19 |
| Stakeholders | 6 | 75 | 6 | 60 | 5 | 83 | NA | | | | | | 17 | 12 |
| Professionals at borders | 2 | 25 | 4 | 40 | 1 | 17 | | | | | | | 7 | 5 |
| **Gender** | | | | | | | | | | | | | | |
| Women | 4 | 50 | 1 | 10 | 3 | 50 | 49 | 78 | 19 | 63 | 19 | 86 | 95 | 68 |
| Men | 4 | 50 | 9 | 90 | 3 | 50 | 14 | 22 | 11 | 37 | 2 | 9 | 43 | 31 |
| Transgender | 0 | 0 | 0 | 0 | 0 | 0 | 0 | 0 | 0 | 0 | 1 | 5 | 1 | 1 |
| **Age (years)** | | | | | | | | | | | | | | |
| 18–30 | 0 | 0 | 1 | 10 | 1 | 17 | 15 | 24 | 6 | 20 | 9 | 41 | 32 | 23 |
| 31–45 | 4 | 50 | 4 | 40 | 3 | 50 | 34 | 54 | 14 | 47 | 8 | 36 | 67 | 48 |
| >45 | 4 | 50 | 5 | 50 | 2 | 33 | 14 | 22 | 10 | 33 | 5 | 23 | 40 | 29 |
| **Education** | | | | | | | | | | | | | | |
| None | 0 | 0 | 0 | 0 | 0 | 0 | 0 | 0 | 2 | 7 | 0 | 0 | 2 | 1 |
| Primary | 0 | 0 | 0 | 0 | 0 | 0 | 6 | 1 | 8 | 27 | 1 | 5 | 15 | 11 |
| Secondary | 0 | 0 | 0 | 0 | 0 | 0 | 15 | 24 | 5 | 17 | 6 | 27 | 26 | 19 |
| Higher/university | 8 | 10 | 10 | 10 | 6 | 10 | 42 | 67 | 15 | 5 | 15 | 68 | 96 | 69 |
| **Professional activity** | | | | | | | | | | | | | | |
| None | 0 | 0 | 0 | 0 | 0 | 0 | 21 | 33 | 10 | 33 | 2 | 9 | 33 | 24 |
| Part-time | 0 | 0 | 0 | 0 | 0 | 0 | 10 | 16 | 1 | 3 | 4 | 18 | 15 | 11 |
| Full-time | 8 | 10 | 10 | 10 | 6 | 10 | 32 | 51 | 19 | 63 | 16 | 73 | 91 | 65 |
| **Civil status** | | | | | | | | | | | | | | |
| Single/widowed | 3 | 38 | 2 | 2 | 1 | 17 | 14 | 22 | 3 | 1 | 3 | 14 | 26 | 19 |
| Married/relationship | 4 | 5 | 8 | 8 | 5 | 83 | 43 | 68 | 27 | 9 | 19 | 86 | 106 | 76 |
| Divorced/separated | 1 | 13 | 0 | 0 | 0 | 0 | 6 | 1 | 0 | 0 | 0 | 0 | 7 | 5 |
| **Number of children** | | | | | | | | | | | | | | |
| None | 3 | 38 | 2 | 2 | 2 | 33 | 6 | 1 | 2 | 7 | 3 | 14 | 18 | 13 |
| 1–2 | 3 | 38 | 3 | 3 | 2 | 33 | 38 | 6 | 10 | 33 | 14 | 64 | 70 | 5 |
| ≥3 | 2 | 25 | 5 | 5 | 2 | 33 | 19 | 3 | 18 | 6 | 5 | 23 | 51 | 37 |

NA: not applicable.

caused by the disease, such as "*hot-cold sensations*," "*chills*," "*fever*," "*headache*," and "*body ache*." Many participants reported having experienced malaria.

"*Living conditions*" also emerged from the conversations, especially among participants living close to places with "*mosquito proliferation*," such as factories, coastal settlements and/or near to trash, e.g., as noted by a pregnant woman in Indonesia:

"*[There is] an abundance of trash around our coastal habitation, giving rise to the proliferation of mosquito breeding spots*." (East Nusa Tenggara, Indonesia, 25 years, woman).

"*Children*," "*pregnant women*," "*older people*," and "*travelers*" were identified as being at higher risk of contracting malaria than the general population; this was particularly mentioned in Rwanda. In Peru, some participants emphasized the importance of diagnosing malaria during pregnancy.

Most participants in Peru recognized two different parasites, *P. vivax* and *P. falciparum*, and their related treatments. Some misconceptions were also reported, such as malaria being caused by "*a virus transmitted by mosquitoes*" and the "*virus staying in the blood.*"

Some participants shared their strategy to prevent malaria, such as seeking routine blood checks, especially when feeling sick, as stated by a woman caregiver in Indonesia:

"*Every time I'm sick, I always check my blood for malaria.*" (Papua, Indonesia, caregiver, 32 years, woman).

## Perceived advantages of the new devices

**What is their added value?.**  In all three countries, the key advantages mentioned regarding the new devices were the non-invasiveness and comfort (no pain), especially when compared with conventional methods, such as blood specimen collection using needles and "*finger prick*," which were considered "*painful*," especially for children. In Peru, campaigns to check for asymptomatic cases are conducted every 3–4 months, including in children:

"*Children are terrified of needles! Children even hide and do not want to come [to school] because they know they are going to be pricked.* (Peru, teacher, 54 years, woman).

"*Sometimes people are afraid of getting pierced because of the pain. But these methods of breathing, scanning the skin, would be better, since one gets results without the body harmed in any way.*" (Rwanda, caregiver, 30 years, man).

Their simplicity, ease of use, and rapid diagnostic capabilities enabling real-time disease diagnosis were perceived as particularly beneficial in remote areas with limited healthcare infrastructure. In all three countries, the ability to carry the devices anywhere was seen as a game-changer for populations living in remote areas:

"*I think this tool [skin scan or VOC] is excellent, especially for those of us who work or live in remote areas. Because it doesn't require blood sampling, no need for a microscope, no need for reagents. Just a scan, or just blowing.*" (Papua, Indonesia, medical expert, 43 years, woman).

The time-saving factor, reducing HCWs' workload, and allowing service recipients to rapidly receive their results, along with the devices' straightforward operation that required minimal training, i.e., they were not dependent on specialized staff, were perceived as pivotal benefits, particularly among HCWs:

"*I can see that this tool [hand scan] will test more people in less time. People won't be waiting longer, and the lab technicians won't have a high workload.*" (Rwanda, HCW, 32 years, woman).

"*If these tools are in the community, trained people can help others.*" (Rwanda, stakeholder, 38 years, man).

The availability of testing opportunities without requiring healthcare assistance further reinforced the advantages of these tools, as testing kits could be used in schools, enabling rapid and convenient monitoring of health conditions.

Among the four types of devices, the skin scan (renamed the "*UV tool*" or "*scanner*" in Rwanda) and the skin odor detector were the preferred tools among various end-users, experts, and community members. In all countries, these two tools were regarded as being particularly easy to use and user-friendly, especially for children under 5 years and pregnant

women. They were deemed suitable for use both by individuals and for mass screening activities, with specific mention made of their usefulness for the early detection and treatment of malaria to prevent malaria complications and the misuse of medications.

"*Taking anti-malaria treatment without being tested is common, and I think with these new tools, people should take medication after being tested.*" (Rwanda, professional at a border, 50 years, man).

The ease of transportability to remote areas, coupled with their minimal requirement for new supplies and lack of waste generation (no trash) were perceived to be additional advantages. These devices could be carried by community health workers, and they could be stored at community leaders' homes or small stores:

"*I pick the first one [skin scan], because you carry it. You put it in your pocket, and you go anywhere. But the other two that you mentioned [breath test, saliva test], you had to carry your personal protection equipment, your glove box, more luggage, but this [skin scan] is something very practical. Just like the cell phone they put it here [vest pocket], so I can put it here and…*" (Peru, HCW, 27 years, woman).

In Indonesia, the comfort and lack of pain associated with saliva-based tests were viewed as important advantages (potentially alleviating the fear of needles):

"*If it's from blood, it might be painful. That's the advantage. This is like the HIV test already using saliva. If using saliva, there's no need... no pain. If children are asked to produce saliva, they might prefer it compared to having to be pricked with a needle to take out blood. Children won't be scared. So, this is the advantage.*" (Indonesia, doctor, 56 years, man).

Participants from Rwanda and Peru also expressed positive attitudes toward saliva-based tests, due to their familiarity with these types of tests for other uses, such as rapid COVID-19, pregnancy, and HIV tests:

"*It's similar to the COVID-19 test.*" (Peru, caregiver, 38 years, man).

**Concerns regarding the new devices**

Various concerns that could potentially affect the use of the tests emerged from the discussions. These concerns included parasite/species detection, reagents needed, factors influencing test outcomes (such as hygiene, drugs, and diseases), cultural barriers, and contextual challenges.

**Which malaria species can these devices detect?.** The inability of all the types of devices to detect which malaria parasites/species are present, potentially leading to the need for additional tests and/or unnecessary and inappropriate medication, was a major concern in all countries, as summarized by a doctor in Papua, Indonesia:

"*In Papua, there are five types of malaria, and the treatments are different. For tropical malaria, primaquine is only taken for 1 day. For patients with tertian malaria, it's 14 days. For pregnant women, we don't give primaquine. So, if these tools can't detect whether it's tropical or tertian malaria, it's a bit confusing because how will we decide on the treatment?*" (Papua, Indonesia, doctor, 43 years, woman).

**Could the results be biased?.** "*Body odor,*" such as from sweating or the use of perfume, raised uncertainties about the reliability of the skin scan test, referred to by some participants as the "*smell malaria*" device. In Indonesia, this issue

was raised by doctors concerned about villagers who might not bathe when they are unwell. In Peru, this concern was raised by all participants:

> "*If I use perfume, will it come out correctly?*" (Peru, Iquitos community member, 41 years, woman).

> "*Here, it's usually quite hot, perhaps you could mistake odor for sweat. We could miss the diagnosis.*" (Peru, stakeholder, 44 years, woman).

Factors that could affect the breath of an individual, such as alcohol, strong-smelling foods or drinks, or poor oral health, and could potentially lead to inconclusive results were specifically highlighted by participants in Rwanda and Peru:

> "*There are different factors that affect the breath. For example, if a person has had alcohol or has smoked, eats chewing gum, any kind of food there's always going to be different types of odors!*" (Peru, HCW, 30 years, man).

The exhalation device also raised concerns about hygiene among some participants, who felt it was unpleasant to blow into a device that had already been used by others. The idea of "*sharing breath*" was particularly uncomfortable for informants in Peru.

Some participants felt that the results of the skin scan might also be influenced by factors that can affect blood color, e.g., anemia, dengue, or the presence of medication such as quinine:

> "*The device measures the color of the blood, but there are multiple factors that alter the color of the blood. It's not only malaria, but it can also be, for example, anemia, dengue. The presence of bacteria.*" (Peru, HCW, 30 years, man).

Additionally, some participants questioned whether the skin scan had already been validated with different ethnic and racial groups:

> "*It's one thing to test it in Caucasian population, another black population, and another in Hispanic population!*" (Peru, professional at a border, 36 years, woman).

Concerns regarding the need for reagents for the saliva-based test, including their expiration dates and temperature stability, were raised in Indonesia and Peru. This issue was considered to add a layer of complexity to the practical implementation of this tool:

> "*The saliva requires its own reagent, and we also have to consider whether the reagent has an expiration date or not.*" (Papua, Indonesia, medical expert, 43 years, woman).

Some indigenous community members and experts reported cultural barriers related to two devices: the exhalation device in Peru and the skin scan in Rwanda. The indigenous populations in Peru might react negatively to the exhalation device, as it could be perceived to be "*stealing their soul*":

> "*In the community, you don't know if through this test you 'steal their soul'.*" (Peru, stakeholder, 60 years, man).

Contextual logistical challenges also emerged. The over-reliance and dependence on technology for the skin-scan and VOC devices, such as smartphones and laptops, were seen as introducing new challenges, especially in Indonesia. The potential for an inconsistent energy supply due to low batteries was noted to be a drawback that could affect the accuracy of the results.

*"If the battery is running low, the energy can become inconsistent. It's like when we use digital blood pressure monitors, and if the battery is low, the accuracy may be affected."* (Indonesia, stakeholder, 44 years, woman).

The disposable supplies and the need for replacements, such as the mouthpieces required for the saliva-based and exhalation devices, were identified as major drawbacks because of the trash generated. This could become an environmental issue, as noted by a teacher:

*"It's important to look at environmental issues as well. In the device where you blow, a lot of straws are wasted and it generates garbage."* (Peru, teacher, 34 years, man).

This concern was also raised by health professionals, who expressed that the use of saliva-based and exhalation devices would necessitate protective equipment, consequently generating additional waste that may be disposed of inappropriately in the environment.

Ensuring the correct distance between a skin scan device and the skin of an individual being tested was often questioned by respondents from Peru:

*"The other one can come close to me and suddenly I don't get a good reading because of the margin of distance or a child can move."* (Peru, HCW, 38 years, woman).

**Are the new devices safe?.** In Rwanda, HCWs and care recipients alike were concerned about their safety and potential harms when using the skin scan, particularly regarding exposure to ultraviolet radiation. They indicated a preference for the other types of devices:

*"There are some methods that I can accept and others that I can't, such as testing using the skin scan, where it uses UV rays that penetrate the human skin. But for other tools, I won't hesitate to use them, given the provided explanations regarding their safety about my health because they don't harm my body, and they give results instantly."* (Rwanda, HCW, 40 years, man).

Some HCWs even suggested that some people might think that exposure to the skin scan could reduce their life expectancy:

*"Regarding the scanner, people would believe that, if someone is scanned in one way or another on a certain body part, the life expectancy would reduce. Thus, if I would have to live like 100 years, it would be reduced to 80 or 90 years instead. That's what most people would have in their mindsets."* (Rwanda, HCW, 32 years, woman).

The use of the skin scan generated the opposite feelings among participants in Peru. They expressed confidence regarding the skin scan device because it has direct contact with the skin and can therefore detect "*something*" in the blood:

*"I trust anything to do with blood more than breath. Blood is stronger, in blood you can see everything and you can measure more."* (Peru, teacher, 61 years, woman).

Similarly, the saliva-based test, which uses a bodily fluid, was perceived as being more reliable than tests that do not use bodily fluids or make direct contact with the body:

*"I think it's better because it's saliva and it comes directly from the body, it gives you confidence."* (Peru, pregnant woman, 37 years, woman).

In Rwanda and Peru, concerns about the potential for transmission of respiratory diseases, such as tuberculosis (TB) or even COVID-19, from the "*spit in the air*" were raised:

"*But this is the way that I think can be a source of contamination of diseases such as hepatitis, tuberculosis.*" (Rwanda, HCW, 40 years, man).

Most healthcare providers and professionals working at borders in Peru found the saliva-based test "*disgusting*" and "*unsafe.*" They expressed a need for protective equipment if they were to be around individuals spitting in front of them, often related to the risk of TB transmission:

"*We would have to be protected somehow. We're talking about bodily fluids. Besides malaria, in the indigenous area there's tuberculosis. So, we would need adequate protection. So we would have to be protected and the population would have to be protected as well. […]. The mouthpiece is individual but if a person has TB and they leave microbes, not in the mouthpiece but in the device, inside?*" (Peru, professional at a border, 36 years, woman).

"*Very risky. With a TB patient we can become infected. You contaminate the environment, you contaminate the health personnel, it is a little complicated.*" (Peru, HCW, 36 years, woman).

In Peru, HCWs generally disagreed with the use of the saliva-based device in a school "*full of children spitting,*" which would require an increase in the supply of protective gear needed for each individual. Community members, parents, and pregnant women did not react in the same way as the HCWs and professionals at borders but found the device "*less practical,*" "*harder to quickly be used for detection,*" and with "*a longer waiting time for the results.*"

**What do participants think about non-healthcare professionals knowing their malaria test results?.** When asked about information and the disclosure of one's malaria status, in the context of the possibility of non-health professionals using the new tools, opinions varied among the three countries. In Peru, participants perceived that there was no stigma attached to malaria; therefore, their malaria status could be known and disclosed by any trained individuals.

"*There is no stigma, malaria can happen to anyone.*" (Peru, caregiver, 30 years, woman).

Participants in Rwanda insisted on the confidentiality of information regarding malaria testing and results, as noted by a young woman working at the border:

"*The perso n who tests a patient should be the one to disclose the results, no one else.*" (Rwanda, professional at a border, 22 years, woman).

Some respondents in Peru feared that individuals who worked in coca or cocaine production and/or trafficking illegal drugs might believe that the skin scan could read their fingerprints and identify them and that this information could then be used by police officers. In this case, concerns around confidentiality were not related to an individual's malaria status but to their identity, which could then be used for purposes other than medical ones.

"*The 'fingerprint' thing could be problematic [skin scan]. One, because it's infrared. They might think it is systematic, something computerized, or that they are collecting their information. Worse, in cocalero communities [communities that grow coca for cocaine production], there could be resistance there.*" (Peru, professional at a border, 36 years, woman).

**Implementation of the new devices**

The discussions also addressed the implementation of the devices, exploring questions such as where to integrate them (e.g., airports, schools), the optimal times for their use (e.g., only during specific periods, such as the rainy season), the main beneficiaries of these devices (who could benefit most), the entities responsible for their nationwide implementation, and what was specifically needed to implement the devices within their country.

**Where should the new devices be implemented, and who are the potential testing providers?.** In all countries, participants felt that the new tests would better serve the population if implemented in settings that lacked laboratories, equipment, or health professionals, such as remote areas, impoverished regions, malaria-endemic areas where a shortage of HCWs exists, and difficult-to-reach communities.

"*In endemic areas and in remote areas, far from health facilities that have the capacity for microscopic diagnosis. So, that's where we would use these tools.*" (Peru, health authority, 60 years, man).

Specific settings/places where populations are active and mobile were also identified by the participants as potentially useful settings for the introduction of the tests. These included borders, schools, churches, healthcare centers (that lacked laboratories), and airports; challenging environments, such as mining areas, forests, mountains, and palm oil plantations; social spaces, e.g., recreational areas, playgrounds, sports facilities, multipurpose halls, bus terminals, markets, and community health centers; military bases near borders; and NGOs assisting refugees and communities in need who live near borders. Some participants also mentioned that they would like to have these devices at home.

In Peru, participants recommended the use of the skin scan and VOC tests for rapid testing in schools and neighborhoods as part of routine screening in malaria-endemic areas, to be followed up with a confirmatory test in the case of a positive result:

"*It would be two plans. Plan A, if I don't have the resources to bring someone with me to do a thick smear immediately, because we have to start treatment immediately, right? We could treat both strains and then do follow-up. But if it's possible, I would take both, the equipment to discriminate and to determine if it's falciparum or vivax. And on that, define my treatment.*" (Peru, health authority, 43 years, man).

In Indonesia, some respondents suggested that these devices could also be integrated in evacuation centers during disasters or in areas currently undergoing infrastructure development (e.g., new road construction sites).

"*I would prefer this tool to be used in remote areas, for example, in evacuation centers during disasters, like when there was a flash flood in Sentani here, and people stayed in evacuation centers for about a month. Even though it's close to the city, when we're in evacuation, it means we have to prepare tools in case someone gets sick. Or in rural areas, in the forest, on the mountain, something like that.*" (Indonesia, stakeholder, 43 years, woman).

Participants in Rwanda noted that the choices of location and device would depend on the risk of infection transmission. They highlighted that the skin scan tool would be most suitable for settings other than health facilities, as the saliva- and exhalation-based devices might pose a risk of disease transmission:

"*There is a method that does not cause any problem, the one we were talking about, using infrared, although maybe we were afraid of having a problem of the skin. But this is the way that I think that the skin scan is allowed to go outside the health institutions, because all other methods like those of breathing in the vapor test kit, can be a source of contamination of diseases such as hepatitis and tuberculosis, and should be used in institutions of health.*" (Rwanda, HCW, 40 years, man).

Airports were often highlighted with reference to the COVID-19 pandemic, during which time screening was performed at airports in some countries. Malaria screening at airports was seen as a useful early detection measure that could prevent the (re)emergence of malaria by controlling its spread, especially for new arrivals:

"*It's interesting. If there's a scan, perhaps...at the airport, similar to COVID-19. Hand sanitizer is given to everyone upon entry. Just that could work. Well, for instance, in places like Labuan Bajo, where there's no malaria. For new arrivals, they should be checked to avoid...if it becomes prevalent, it will be hard to minimize it again. Early detection, I think, will be used if it has good accuracy.*" (Indonesia, doctor, 56 years, man).

Settings close to bodies of water and locations where sailing and fishing take place were also considered to be high-risk sites for malaria infection due to the presence of mosquitoes, as mentioned by a community member from Rwanda:

"*The place where the tools can mostly be important for people working with water bodies, such as fishing, sailing, because it is in water bodies where mosquitoes breed. So, testing these people in permanent contact with water is a good initiative.*" (Rwanda, community member, 26 years, woman).

Most respondents did not see the need to integrate the tests in healthcare centers located in cities, as they were considered to be well-equipped, with laboratories that could easily perform blood tests, particularly in Indonesia:

"*In the city, we can still use a tool with 100% accuracy, especially those using blood sampling. I would prefer this tool [skin scan or VOC] to be used in remote areas because in Papua, many people live in the middle of the forest.*" (Papua, Indonesia, doctor, 43 years, woman).

In Indonesia, participants highlighted the importance of primary health centers, that offer a wide range of basic healthcare services, community health centers that serve as the first point of contact for most, and. Health cadres with authorized volunteers to administer malaria medication, especially in remote areas.

"*If the device is available, and if it's already been through trials, it could be used in hospitals, emergency departments, or community health centers (*Puskesmas*).*" (Indonesia, stakeholder, 46 years, man).

Some participants referred to the tests as being useful for the implementation of "*mobile testing,*" especially for remote areas and for mass screening.

Due to their ease of use, potential providers identified included HCWs who had received minimal training and were not skilled in laboratory technologies, community health cadres (in Indonesia), community HCWs within communities (Rwanda), and professionals working at border areas.

In Peru, participants agreed that key community partners, including teachers and the owners of small shops, could be trained to use the tests if needed in certain situations, e.g., after hours (when health facilities are closed).

In Indonesia, the tools were seen by doctors as beneficial for travelers, especially if they could self-check their status before entering malaria-prone regions. Teachers also mentioned that staff from *Puskesmas* could visit schools to conduct follow-up if positive results occurred.

In Rwanda, some participants highlighted the necessity of making the tools accessible for the testing of individuals in occupations with a high risk of malaria infection, such as security officers and those working near bodies of water such as marshlands.

However, some health experts raised concerns regarding the risk of incorrect usage of any of the types of devices if used outside of healthcare facilities/services and not under the supervision of a healthcare professional. If the tools

were to be implemented outside health facilities, they recommended seeking confirmation and further treatment at health facilities.

Some Indonesian HCWs stressed the importance of combining public and professional use, foreseeing a scenario where workers (who were not health workers), especially in malaria-prone locations, could benefit from routine monitoring to prevent delays in diagnosis.

**When to implement the tests and who would/could benefit the most?.** To the question "*Do you think there is a specific time to use this tool/these tools or can it/they be used anytime?*", the majority of the participants suggested using them more frequently during the rainy or harvest seasons when malaria is more prevalent due to increased mosquito activity and the associated increased risk of transmission.

"*Malaria is likely to affect people during rainy season, and people use to confuse it with flu. I think it will be good if they use the tools for malaria testing during rainy seasons.*" (Rwanda, community member, 22 years, woman).

Some participants suggested regular monthly testing for pregnant women and routine screening for children aged between 1 and 5 years.

Many participants pointed out that the individuals who could benefit the most from these devices were those who lived in difficult-to-reach communities, **i.e.**, individuals who faced challenges accessing healthcare services or those unable to frequently visit health facilities, such as those who live in remote villages.

Most respondents also noted that those individuals who are at the highest risk of malaria could benefit the most from these devices. These included pregnant women, children under five, individuals with disabilities, older people, people attending mass gatherings, outdoor workers such as security personnel and motorcyclists who spend prolonged periods outside, as well as individuals with limited resources:

"There are other people who would benefit the most from the use of these methods. Those are the people with low financial capacity, as they are usually afraid of wasting time by attending health facilities for malaria test. As long as there is an easier and faster way of malaria testing, they will go for it and know their health status. In statistics it is obvious that communities with low financial capacity are also more vulnerable to severe malaria. If there are those methods of testing them regularly and easily, it will help to reduce the risk of severe malaria cases or deaths." (Rwanda, teacher, 32 years, man).

Age was considered to be an important characteristic by parents, teachers, and HCWs in Peru and Rwanda. They felt that tools that are easy to use would be especially relevant for screening young children:

"*A 3-month-old infant can't be told to spit or breathe in a kit, and if we want to test the baby, we must have the specific test like the skin scan*." (Rwanda, HCW, 59 years, man).

**Who should be in charge of implementing the rollout of these tests?.** In all three countries, their respective Ministries of Health were regarded as being the entities responsible for implementing programs to support the use of the devices. Participants in Indonesia and Rwanda also mentioned the importance of collaboration with other institutions, such as the Ministry of Education, the Ministry of Local Government, the Food and Drug Authority (in Indonesia), and other malaria partners/institutions, as highlighted by a stakeholder:

"*Three institutions have to work hand in hand, these are the Ministry of Health, Ministry of Local Government, and Ministry of Education. But the coordination and lead should be dedicated to the Ministry of Health. I am also thinking about church leaders and other institutions, including those in charge of security, to support the implementation*." (Rwanda, stakeholder, 38 years, man).

In Peru, health professionals and authorities highlighted that the Ministry of Health would look to international health organizations, e.g., the World Health Organization and the Pan American Health Organization, for clear recommendations on their use.

**What is needed to implement these tests?.** We asked participants for their suggestions and recommendations related to the devices and their implementation. While their answers focused mainly on the technological dimension, some responses also touched on the "human dimension."

Regarding the technology, all participants highlighted the importance of increasing the capability of the devices to identify different species of malaria parasites. They also noted the importance of providing clear and easy-to-understand instructions for users, including how to correctly maintain and store the devices and the reagents (for the saliva-based test). This was particularly highlighted by participants in Indonesia, who identified challenges associated with the use of tests in settings other than healthcare facilities and storage at the correct temperature (e.g., storage in non-airconditioned rooms, possible losses due to mishandling, and broken devices). These challenges would need to be mitigated by setting out appropriate processes in any new policies.

According to respondents in Rwanda, all device types must respond to three important characteristics, which are being "*affordable*," "*accessible*," and "*available*":

"My suggestion is to make the tools affordable and accessible, deploy them at village level, and do not make them hard to reach. You can deploy them in district pharmacies as they are the ones supplying health centers. When they will decide to bring those tools, they should bring enough quantity so that people can access them at the level where every household has at least one." (Rwanda, stakeholder, 38 years, man).

One doctor in Indonesia suggested it would be useful to provide a printout of the results for further referral to a health facility:

"*Will [the device] be provided with something like that [print out]? So the patient has a proof of the test, so that we can have a document to bring during their referral to get a confirmatory test. We can counsel them and say you are malaria-positive and bring this to the nearest health facility, for those in public places.*" (East Nusa Tenggara, Indonesia, doctor, 48 years, woman).

In Peru, health authorities recommended adding the possibility of integrating some type of data component that linked to an electronic database and would allow healthcare professionals to be able to input a patient's name and basic information that could be linked to their results. This would be useful for monitoring malaria cases:

"*If you have a memory system, so that at the end of the day I can go through the data and pull out a summary. So, I know, for example, how many I evaluated that day, how many positives there were, how many inconclusive [had indeterminate results].*" (Peru, stakeholder, 44 years, woman).

In Rwanda, participants recommended the devices be integrated as complementary diagnostics into the existing system but not as a replacement for actual malaria screening and testing methods, such as microscopy, to ensure different malaria parasites could be identified:

"*This will come to support what is already in place for malaria testing. If adopted, they should be used as additional tools.*" (Rwanda, professional at a border, 51 years, man).

Peruvian health professionals and authorities recommended the integration of a long-lasting battery, possibly a rechargeable one, and that efforts should be made to develop a hybrid option for remote areas, where no electricity is available.

In all countries, participants highlighted the need to raise awareness of the devices among communities and gain their acceptance, by organizing extensive public information and, community mobilization campaigns. For example, in Peru, it was mentioned that sometimes something "new" might initially be met with mistrust from the community, and there might be a need for extensive public information.

Socialization of parents about malaria and the new devices was particularly highlighted in Indonesia, as participants perceived that some parents may be opposed to the initiative, complicating the gaining of consent for testing.

In Rwanda, training of staff was deemed essential so they could integrate the devices into their routine:

"*Trainings are needed at all levels. Community health workers can also provide training where there is a need.*" (Rwanda, professional at a border, 50 years, man).

**What cost is acceptable?.** In Indonesia, the cost of all the types of devices was identified as a potential drawback that could impact their adoption and implementation. In Peru, the cost of the saliva-based test was compared with that of a blood-based RDT, used by the malaria program and capable of differentiating *P. vivax* and *P. falciparum*. The blood-based test costs US\$1–2 per test versus an estimated \$5 for the saliva-based test. In addition to being more expensive, the burden of needing to perform an additional confirmatory test was considered to be a negative aspect of the new tests. Hence, authorities might be reluctant to invest in the saliva-based test, as noted by a stakeholder:

"*You have to do it twice! You are going to take the saliva sample, have him wait 20 minutes, and then again, if it comes back positive, do an additional rapid test. So, it's already twice, it creates more difficulty for us.*" (Peru, health authority, 57 years, man).

Conversely, in Rwanda and Peru, the skin scan and VOC devices were perceived as only requiring one initial investment that participants believed would then, in turn, pay for itself (and reduce the time required) by reducing the number of thick smears that would need to be performed during screening campaigns. The cost of test consumables, considered to be expensive, would be reduced, as there would no longer be a need for associated supplies, as highlighted by an HCW in Rwanda:

"*The other benefit is for health facilities, as there are consumables that were used to test malaria which will no longer be needed. Among those consumables, so many are highly expensive. This is the reason why I was wondering if these new methods will be used in parallel with the existing ones, or if they will be a replacement. If we stop using the consumables, the funds that were spent on them will be saved. Generally, the country will benefit, through the finance, because the people will not frequently be sick.*" (Rwanda, HCW, 59 years, man).

**What would be the impact on malaria prevention?.** Finally, the impact the new tests might have on malaria prevention arose spontaneously during the FGDs and SSIs, with participants sharing mixed feelings.

For some participants, the tools could positively impact malaria prevention by increasing mass screening and preventing malaria complications and self-medication. As already mentioned, the ability of the tests to help break the chain of transmission (through early diagnosis) was considered to be a benefit. HCWs in Indonesia highlighted the importance of using RDTs during the malaria elimination phase:

"*Once we achieve elimination, if just one case appears, it's considered an outbreak, and we must treat it immediately to prevent it from reappearing and causing the disease to recur. That's why when one case is found, we immediately examine the radius. We don't want anyone to be already infected and in the incubation period without showing symptoms. It's better for us to work hard at the outset rather than wait for it to spread everywhere. So, during this elimination phase, rapid diagnostic tools are crucial.*" (East Nusa Tenggara, Indonesia, doctor, 56 years, man).

Conversely, some participants feared that preventive measures, such as sleeping under mosquito nets, would stop, a concern particularly raised in Rwanda. Participants also expressed concerns that people might believe that they could easily undergo testing and receive treatment, leading to a lax attitude toward malaria prevention. Additionally, information and knowledge about the new tools could lead to some misconceptions about the cause of malaria. Traditionally associated with blood, the introduction of saliva- or VOC-based tests might confuse individuals regarding malaria transmission:

> "*The disadvantage is that people will be careless regarding malaria prevention measures, because they think it is no longer important to use mosquito nets or adding chemicals to improve their effectiveness, as they will go on getting tested for malaria. Thus, I think that there will be carelessness with people, believing that it is easy to get tested for malaria and the treatment is faster than it used to be. I think that it is dangerous that the people will no longer value malaria prevention measures.*" (Rwanda, teacher, 40 years, woman).

A synthesis of the findings from the focus group discussions and the semi-structured interviews in Indonesia, Peru, and Rwanda is shown in Fig 2, highlighting the perceived advantages and disadvantages of the new non-invasive tools compared to traditional blood sampling methods.

## Discussion

This qualitative study is one of the few studies to explore the perceptions of various end-users regarding non-invasive malaria diagnostics and expands on previously published work that examined national stakeholders' perceived value of these diagnostics [26,27]. This research contributes innovative evidence and knowledge in this field, from three geographically diverse countries on three continents.

Participants from all three countries identified several advantages of the non-invasive malaria testing tools. The devices are perceived to be simple to use and user-friendly, to require minimal training and equipment, to reduce diagnosis time, and to be less painful than blood tests. These benefits could potentially improve screening coverage, as noted in other studies [33]. The introduction of these tools in remote areas or in settings that lack laboratory equipment and infrastructure was perceived to be an added benefit in the fight against malaria, as they could increase the number of people screened and reduce the dependence on health centers. This finding is in line with that from a survey conducted among national stakeholders involved in malaria control, 94% of whom felt that non-invasive malaria diagnostics could greatly enhance the provision of malaria care in difficult-to-reach areas [26]. The potential to screen individuals outside of healthcare settings, such as in airports or schools, by non-healthcare providers was also perceived to be a major benefit. In malaria-endemic sites, the possibility of routinely screening all individuals with a rapid, non-invasive device and then referring any individuals who have a positive result for a confirmatory microscopy test was seen as a major time-saver and would reduce community "fears" regarding the taking of finger prick samples.

Disadvantages of the non-invasive malaria diagnostic tests reported by participants were the inability to differentiate between parasite species, the safety of the devices, and potential stockouts of consumables and reagents [33]. Participants expressed that these new devices need to be discussed with consideration of their potential environmental consequences. While some participants highlighted the sustainability of the tools, emphasizing trash and waste reduction, others suggested the opposite and feared that one-time-use devices would generate more trash. Participants also emphasized contextual logistical challenges, such as unreliable supply chains and transportation difficulties to remote areas where malaria is a major public health threat. Some participants expressed fears and doubts about potential factors that could affect the test results with these new devices, such as the skin color of test recipients, reagent availability, hygiene, and the impact of medicine or alcohol consumption. Issues around the disclosure of individuals' malaria status appeared to be context-specific, as it was reported to be sensitive information in Rwanda and Indonesia but not in Peru and was linked to stigma in Rwanda and Indonesia. The risk of transmitting other diseases, such as TB, while using the devices

| Perceived advantages of the new types of tests (compared with those that use blood specimens) | Skin scan | Saliva-based RDT | Exhaled breath (VOC) | Skin odors (VOC) |
|---|---|---|---|---|
| | | | | |
| Comfort – no pain | x | x | x | x |
| Easy to use | x | x | x | x |
| User-friendly | x | x | x | x |
| Rapid results | x | x | x | x |
| Time-saving: reduce laboratory and HCW workload | x | x | x | x |
| Good for remote areas | x | x | x | x |
| Easy to transport | x | x | x | x |
| Minimal training needed | x | x | x | x |
| Useful for mass screening | x | | | x |
| Familiarity with this type of test (e.g., for pregnancy, COVID-19) | | x | | |
| **Perceived disadvantages (compared with blood specimens)** | | | | |
| Cannot distinguish between malaria parasite species | x | x | x | x |
| Not safe (UV radiation; limit life expectancy) | x | | | |
| Affected by body odor | x | | | x |
| Affected by skin color (ethnicity) | x | | | x |
| Issue around oral hygiene/breath | | | x | |
| Diseases or drugs could affect blood color | x | | | |
| Reagents (stock, expiry date, maintenance) | | x | | |
| Difficulty maintaining correct distance from skin | x | | | |
| Technology dependence (battery) | x | | x | x |
| Generate trash | | x | x | |
| Not safe: infection risk, disease transmission | | x | x | |
| Perceptions that malaria can only be detected in blood | x | x | x | x |
| Body hygiene | | | | x |
| Cultural barriers | x | | x | |

**Fig 2. Synthesis of findings from the focus group discussions and semi-structured interviews in Indonesia, Peru, and Rwanda.**

was discussed. Cultural beliefs were also seen as potential barriers to the implementation of certain types of devices. The perceptions of malaria among the participants were framed through their experiences of the disease [34]. The belief that

malaria could "only" be accurately detected in blood specimens remained strong. Participants frequently noted the association between the disease, the environment, and living conditions.

We asked participants where they could imagine these new non-invasive devices being deployed and which target population would gain the most benefit from their use. A large majority of respondents indicated a heightened need for these tools during the rainy and harvest seasons, correlating with increased mosquito activity and malaria transmission risk. The skin scan and skin VOC device were seen as being particularly useful for routine screening in schools, community health centers, and neighborhoods within malaria-endemic areas. If the devices proved to be effective at detecting low levels of parasitemia, their use in routine screening could greatly enhance the identification of individuals with asymptomatic malaria who, despite showing no clinical symptoms, can still transmit the malaria parasite to mosquitoes, thereby perpetuating the disease cycle. Early detection and treatment of such cases might be pivotal for controlling the spread of the parasite, especially in countries reaching the malaria elimination phase. This has been demonstrated previously, in a study of a mass screening and treatment approach [35].

In our study, the usability of these non-invasive tools in difficult-to-reach communities, such as remote villages with limited access to healthcare, was also underscored. Individuals living in such areas often face challenges in making regular visits to health facilities, so easy and rapid testing methods are essential. Participants acknowledged the value of these tools for regular testing of pregnant women and routine screening of children under five, given these groups' vulnerability to severe malaria complications. Other high-risk groups identified for whom these tests may be useful included older people, people with disabilities, and outdoor workers.

Currently, these devices are considered to be potentially acceptable as complementary diagnostic tools rather than replacements for existing malaria diagnostics. Participants indicated a preference for using them in remote areas where health services lack adequate equipment and laboratory facilities. Consequently, while non-invasive tools may not replace current malaria tests, they could assist in detecting cases that might otherwise be missed in remote locations and atypical settings, or provide a way to determine which individuals may need a follow-up diagnostic test. By assessing the preferences, concerns, and expectations of potential users from a variety of settings, we gained valuable insights that will help diagnostics developers and manufacturers in shaping the design of these innovative technologies. This is essential to ensure not only are they scientifically and technologically sound but also practical and culturally sensitive. Our study demonstrated that these novel technologies designed to enhance malaria case detection would be widely considered acceptable. Nevertheless, their successful implementation would require community awareness-raising and engagement during the early stages of the development of such devices.

This study had some limitations. The participants were unable to try these new technologies themselves, as they were not available on the market. The responses provided are thus theoretical and may not accurately reflect practical realities. The study was conducted among population subgroups that the researchers considered to be the most relevant to malaria control efforts. However, other subgroups or the wider population might also benefit from non-invasive malaria tools, thus it would also be important for their perceptions to be considered. Future acceptability studies could include a broader representation of communities, civil society, and the general population. A strength of our study lies in the diverse range of participants interviewed, from community members to policymakers, across three countries with distinct contexts.

## Conclusion

The findings of this qualitative study conducted in Indonesia, Peru, and Rwanda emphasize the importance of aligning diagnostic advancements with user expectations to improve the effectiveness and acceptance of malaria control interventions. Engaging various end-users in discussions at the early stages of development and before technology is rolled out will ensure that any new devices are a better fit with the contexts and needs of communities.

At this stage, non-invasive malaria testing tools could be acceptable for use in non-healthcare settings, especially for decreasing the diagnostic gap in remote areas. Participants considered that non-invasive malaria testing tools would be a

useful complementary tool, in combination with currently used blood-based diagnostics, in the fight against malaria. Moreover, this data provides valuable insights that could inform the World Health Organization (WHO) in the development of a Target Product Profile (TPP) for non-invasive malaria diagnostics, ensuring that these tools not only meet global standards but also address the practical needs of end-users.

## Supporting information

**S1 Fig. Visual information sheets used during the FGDs and the SSIs.**
(TIF)

**S1 File. Guide for the SSIs.**
(DOCX)

**S2 File. Guide for the FGDs.**
(DOCX)

**S3 File. Inclusivity questionnaire.**
(DOCX)

## Acknowledgments

We sincerely thank all of the participants who agreed to share their experiences and time with us, the gatekeepers, and the additional parties who helped and supported this study. In Indonesia, we would like to thank the study enumerators, Klinik Agredece, Maumere, East Nusa Tenggara, Yayasan Sorong Sehati, Sorong, West Papua, and Yayasan Kalam Kudus, Jayapura. In Rwanda, we would like to thank the Rwanda Biomedical Center, the Rwanda Immigration Office, and the District Directors of Health where the study was conducted. In Peru, we would like to thank E. Jennifer Ríos Lopez, Alfonso Simoné Vizcarra, and Jhonny Córdova Lopez for their assistance in recruiting, organizing, and documenting the focus group discussions; Marcos Antonio Vasquez Guzmán for improving the sound quality of the audio-recordings prior to transcription; Gabriela Vásquez, Cristina Hidalgo, Nastya Sladkov, and Lizzie Ortiz for quality control of the Spanish transcripts; and Lauren Nussbaum and Emma Ortega for coding and analyzing the data. Finally, we would like to thank our colleague at FIND, M. Seyi Gansallo, the study's Project Manager, and Adam Bodley for medical writing assistance and the editorial support.

## Author contributions

**Conceptualization:** Serafina Calarco, Sonjelle Shilton, Kevin K.A. Tetteh.

**Data curation:** Vanessa Fargnoli, Catherine Thomas, Caroline Thomas, Claudius Novchovick Mone Iye, Valerie A. Paz Soldan, Amy Celeste Morrison, Janvier Serumondo, Ladislas Nshimiyimana, Yvonne Delphine Nsaba Uwera, Elena Marbán-Castro.

**Formal analysis:** Catherine Thomas, Caroline Thomas, Claudius Novchovick Mone Iye, Valerie A. Paz Soldan, Amy Celeste Morrison, Janvier Serumondo, Ladislas Nshimiyimana, Yvonne Delphine Nsaba Uwera, Elena Marbán-Castro.

**Funding acquisition:** Serafina Calarco, Kevin K.A. Tetteh.

**Methodology:** Vanessa Fargnoli, Sonjelle Shilton.

**Supervision:** Vanessa Fargnoli, Serafina Calarco, Catherine Thomas, Caroline Thomas, Valerie A. Paz Soldan, Janvier Serumondo, Sonjelle Shilton.

**Writing – original draft:** Vanessa Fargnoli, Serafina Calarco, Elena Marbán-Castro.

**Writing – review & editing:** Catherine Thomas, Valerie A. Paz Soldan, Amy Celeste Morrison, Sonjelle Shilton, Kevin K.A. Tetteh.

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
