## [Decision Letter · Decision Letter 0]

24 Jun 2025

*“It doesn’t require blood!”*

Dear Dr. Calarco,

Thank you for submitting your manuscript to PLOS ONE. After careful consideration, we feel that it has merit but does not fully meet PLOS ONE’s publication criteria as it currently stands. Therefore, we invite you to submit a revised version of the manuscript that addresses the points raised during the review process.

Reviewer 2 in particular has provided detailed and constructive comments to improve the manuscript. Please respond to these suggestions.

We look forward to receiving your revised manuscript.

Kind regards,

Susan Horton

Academic Editor

PLOS ONE

Journal Requirements:

This study has been funded by UK aid from the UK government (FCDO project number 300341-102), as well as the Swiss Development Cooperation (contract number 81071426).

4. We noted in your submission details that a portion of your manuscript may have been presented or published elsewhere. Please clarify whether this publication was peer-reviewed and formally published. If this work was previously peer-reviewed and published, in the cover letter please provide the reason that this work does not constitute dual publication and should be included in the current manuscript.

5. Please amend the manuscript submission data (via Edit Submission) to include author Claudius Mone, Amy C. Morrison

6. Please amend your authorship list in your manuscript file to include author Claudius Novchovick Mone Iye, Amy Celeste Morrison

7. We notice that your supplementary figures are uploaded with the file type 'Figure'. Please amend the file type to 'Supporting Information'. Please ensure that each Supporting Information file has a legend listed in the manuscript after the references list.

Additional Editor Comments:

The reviewers thought that this paper provides valuable information. Please respond carefully to their comments; reviewer 2 has provided detailed constructive comments.

Reviewers' comments:

Reviewer's Responses to Questions

**Comments to the Author**

1. Is the manuscript technically sound, and do the data support the conclusions?

Reviewer #1: Yes

Reviewer #2: Yes

2. Has the statistical analysis been performed appropriately and rigorously?

Reviewer #1: N/A

Reviewer #2: N/A

3. Have the authors made all data underlying the findings in their manuscript fully available?

Reviewer #1: Yes

Reviewer #2: No

4. Is the manuscript presented in an intelligible fashion and written in standard English?

Reviewer #1: Yes

Reviewer #2: Yes

Reviewer #1: The manuscript is well-written, with correct grammar and proper sentence structure.

The article provides a comprehensive and accurate overview of the current state of malaria diagnostics and the challenges faced. It effectively discusses the limitations of traditional methods and the potential of new technologies that can be used instead. The references are well structured.

The researchers did a good job of explaining complex concepts and providing detailed information about various diagnostic methods. The section on malaria testing tools is informative. It explains the three types of devices, including their current market status and potential applications, which sets up the context for evaluating their acceptability and effectiveness.

Method and material section is well-organized into sections (Study design and sites, Study populations, Sampling procedures, and Malaria testing tools), which help guide the reader through the methodology. Various groups were included (community members, pregnant women, teachers, caregivers, HCWs, border professionals, and stakeholders). This variety supports a rich understanding of different perspectives regarding malaria diagnostics, especially in endemic areas.

Discussion section does an excellent job of capturing a wide range of perspectives on non-invasive malaria diagnostics and demonstrates the potential value of these tools in improving malaria screening and treatment in diverse settings. The text provides a thorough exploration of both the advantages and disadvantages of non-invasive malaria diagnostics. It discusses perceived benefits such as ease of use, reduced pain, and increased screening capabilities as well as potential challenges and limitations such as the inability to differentiate parasite species and logistical concerns.

Reviewer #2: GENERAL REMARKS

In this manuscript, the authors present a relevant qualitative study that aimed to generate evidence from end-users around the potential adoption of non-invasive diagnostic technologies, to determine whether these tools are fit for purpose. To this end, they conducted, in three malaria-endemic countries (Indonesia, Peru, and Rwanda), 24 semi-structured interviews with stakeholders and professionals working at borders, and 16 focus group discussions with community members, caregivers of children under 5 years, pregnant women, healthcare workers, and teachers (total n=139). The participants shared their practical needs, concerns, and acceptability regarding the proposed non-invasive tools. Therefore, the study provides valuable insights and guidance for the development of non-invasive malaria diagnostics, ensuring that these tools meet the needs of end-users. Despite the relevance of this study, the authors should address some minor points before the publication of the manuscript.

MINOR ISSUES

Introduction

• Several sentences lack references (e.g., lines 56, 61, 90, 95, 98, 99, 106). Please carefully review the introduction and add appropriate citations where needed.

• Lines 76–77: Clarify what UNITAID and FIND are.

Materials and Methods

• Line 147: Please explicitly define “stakeholders” (e.g., specify roles like policymakers, NGO representatives, or ministry officials). While contextualized later, a clear definition early would aid readability.

• Line 150: I recommend clarifying the definition of “community members”. Would they be anyone in the community who is not from the other groups (pregnant women, teachers, caregivers of children under five, and healthcare workers)?

• Lines 147–151: Explain the criteria used to establish which method (Semi-Structured Interviews or Focus Group Discussions) was more appropriate for each listed group (e.g., stakeholders, professionals working at borders, community members, pregnant women, teachers).

• Lines 156–163: Justify differences in the endemicity levels and standardize their characterization across the selected areas in the three countries. For instance: Indonesia had no sampling from non-endemic areas, unlike Rwanda and Peru. Peru only sampled highly endemic areas, while Indonesia included low-, medium-, and high-endemicity areas. For Rwanda the endemic areas are mentioned without specifying the endemicity level.

• Lines 156–163: The selected areas in Indonesia and Rwanda lack description as urban or rural, while those in Peru are clearly characterized. I recommend standardizing this classification across all study sites for better comparability.

• Lines 156–163: I suggest to present site characteristics (including FGD/SSI counts and sample sizes) in a table for a better overview of the sampling.

• Line 162: Fix spacing in "and Tumbes".

• Line 164: What is the reason for discussing Peru-specific findings separately and excluding them from this manuscript?

• Line 180: Fix spacing in "aged 18 years".

• Line 252: I recommend sharing SSI and FGD guides as supporting information for more transparency of the applied methods.

• Lines 259–261: Was the information in Figure S1 the only explanation given to participants about the devices? Was the figure translated into local languages? Were the devices limitations or additional details provided?

• I suggest reporting how many people refused to participate or dropped out of the study.

Results

• Explain why according to Table 1 Indonesia had no community members included.

• I recommend the addition of relative values (percentages) and a "Total" column in Table 1 for better interpretation.

• Lines 296–267: The authors mention “the interviewer remaining neutral and unbiased throughout”. Please clarify their stance on answering the questions of participants about the devices. If questions were answered, this should be explicitly stated.

• Figure 2: Correct the misspelling of "malaria" in "Perceptions that malaria can only be detected in blood".

Discussion

• While the study focuses on participants’ perceptions, discussing whether these perceptions align with scientific evidence (especially the perceived disadvantages) would strengthen the paper. The authors are not required to do this modification, but it could add valuable context.

ADDITIONAL POINT

• Regarding data availability: While the authors state that “all data are fully available without restriction”, I was unable to locate the interview transcripts or excerpts of the transcripts relevant to the study in the Supporting Information files.

**Do you want your identity to be public for this peer review?** For information about this choice, including consent withdrawal, please see our Privacy Policy

Reviewer #1: No

Reviewer #2: **Yes: ** Laura Cordeiro Gomes

---

## [Author Response · Author response to Decision Letter 1]

30 Sep 2025

Journal Requirements:

Thank you for the reminder. We reviewed the guidelines and revised the manuscript to ensure it meets the formatting guidelines. Please advise if we missed anything.

Thanks, we uploaded the questionnaire as required

This study has been funded by UK aid from the UK government (FCDO project number 300341-102), as well as the Swiss Development Cooperation (contract number 81071426).

Thanks for your comment, we added the sentence in the cover letter as required.

4. We noted in your submission details that a portion of your manuscript may have been presented or published elsewhere. Please clarify whether this publication was peer-reviewed and formally published. If this work was previously peer-reviewed and published, in the cover letter please provide the reason that this work does not constitute dual publication and should be included in the current manuscript.

Dear Editor, the publication is now formally published (Nussbaum, L., Ortega, E., Ríos López, E.J. et al. Voices from the Amazon: exploring implementor and user perceptions of non-invasive malaria diagnostics in Peru. Malar J 24, 32 (2025). https://doi.org/10.1186/s12936-025-05273-1). As requested, we added a section in the cover letter explaining why this doesn’t constitute dual publication.

5. Please amend the manuscript submission data (via Edit Submission) to include author Claudius Mone, Amy C. Morrison

We have checked the manuscript submission data and both authors Claudius Mone and Amy C. Morrison are listed.

6. Please amend your authorship list in your manuscript file to include author Claudius Novchovick Mone Iye, Amy Celeste Morrison

Thanks, we addressed this change in the manuscript file

7. We notice that your supplementary figures are uploaded with the file type 'Figure'. Please amend the file type to 'Supporting Information'. Please ensure that each Supporting Information file has a legend listed in the manuscript after the references list.

Thank you for bringing this to our attention. The figure has now been correctly uploaded as a Supporting Information file.

Thank you for pointing this out. The Supporting Information captions have been added at the end of the manuscript. We will remove the separate caption file from the submission files, in accordance with the guidelines.

Thank you very much for your comment. We have reviewed the references and none of them seem to have been retracted, please let us know if we missed anything.

Additional Editor Comments:

The reviewers thought that this paper provides valuable information. Please respond carefully to their comments; reviewer 2 has provided detailed constructive comments.

Reviewers' comments:

Reviewer's Responses to Questions

Comments to the Author

1. Is the manuscript technically sound, and do the data support the conclusions?

Reviewer #1: Yes

Reviewer #2: Yes

2. Has the statistical analysis been performed appropriately and rigorously?

Reviewer #1: N/A

Reviewer #2: N/A

3. Have the authors made all data underlying the findings in their manuscript fully available?

Reviewer #1: Yes

Reviewer #2: No

4. Is the manuscript presented in an intelligible fashion and written in standard English?

Reviewer #1: Yes

Reviewer #2: Yes

5. Review Comments to the Author

Reviewer #1: The manuscript is well-written, with correct grammar and proper sentence structure.

The article provides a comprehensive and accurate overview of the current state of malaria diagnostics and the challenges faced. It effectively discusses the limitations of traditional methods and the potential of new technologies that can be used instead. The references are well structured.

The researchers did a good job of explaining complex concepts and providing detailed information about various diagnostic methods. The section on malaria testing tools is informative. It explains the three types of devices, including their current market status and potential applications, which sets up the context for evaluating their acceptability and effectiveness.

Method and material section is well-organized into sections (Study design and sites, Study populations, Sampling procedures, and Malaria testing tools), which help guide the reader through the methodology. Various groups were included (community members, pregnant women, teachers, caregivers, HCWs, border professionals, and stakeholders). This variety supports a rich understanding of different perspectives regarding malaria diagnostics, especially in endemic areas.

Discussion section does an excellent job of capturing a wide range of perspectives on non-invasive malaria diagnostics and demonstrates the potential value of these tools in improving malaria screening and treatment in diverse settings. The text provides a thorough exploration of both the advantages and disadvantages of non-invasive malaria diagnostics. It discusses perceived benefits such as ease of use, reduced pain, and increased screening capabilities as well as potential challenges and limitations such as the inability to differentiate parasite species and logistical concerns.

Reviewer #2: GENERAL REMARKS

In this manuscript, the authors present a relevant qualitative study that aimed to generate evidence from end-users around the potential adoption of non-invasive diagnostic technologies, to determine whether these tools are fit for purpose. To this end, they conducted, in three malaria-endemic countries (Indonesia, Peru, and Rwanda), 24 semi-structured interviews with stakeholders and professionals working at borders, and 16 focus group discussions with community members, caregivers of children under 5 years, pregnant women, healthcare workers, and teachers (total n=139). The participants shared their practical needs, concerns, and acceptability regarding the proposed non-invasive tools. Therefore, the study provides valuable insights and guidance for the development of non-invasive malaria diagnostics, ensuring that these tools meet the needs of end-users. Despite the relevance of this study, the authors should address some minor points before the publication of the manuscript.

- We would like to thank both reviewers for their encouraging and positive feedback on our study and manuscript. We have tried to address the majority of the minor comments raised by Reviewer 2.

MINOR ISSUES

Introduction

• Several sentences lack references (e.g., lines 56, 61, 90, 95, 98, 99, 106). Please carefully review the introduction and add appropriate citations where needed. Thanks for your suggestion, 5 more references have been added.

• Lines 76–77: Clarify what UNITAID and FIND are. Addressed

Materials and Methods

• Line 147: Please explicitly define “stakeholders” (e.g., specify roles like policymakers, NGO representatives, or ministry officials). While contextualized later, a clear definition early would aid readability. Thank you for this suggestion. We had this address, but it was not clear enough. We have included in brackets the specific stakeholders within that sentence (Line 158 to 160).

• Line 150: I recommend clarifying the definition of “community members”. Would they be anyone in the community who is not from the other groups (pregnant women, teachers, caregivers of children under five, and healthcare workers)? Indeed, this needed to be defined better. We added “other” to the term “community members” (Line 163 to 168). This classification included other members of the community who do not fall into the above categories but who are interested in sharing their opinions on the subject. Selection of community members was decided with partners. They could include parents and school staff (recruited at schools), travellers (recruited at travel agencies), established groups at the community (including spiritual/healthcare centres etc.). We have included some wording throughout the manuscript.

• Lines 147–151: Explain the criteria used to establish which method (Semi-Structured Interviews or Focus Group Discussions) was more appropriate for each listed group (e.g., stakeholders, professionals working at borders, community members, pregnant women, teachers). Semi-structured interviews were chosen when the objective was to capture in-depth, individual perspectives on context-specific issues, and when engaging with professionals who might not have the time to participate in a group discussion but could accommodate some time for an individual conversation (e.g., stakeholders, professionals working at borders). In contrast, focus group discussions were used when the aim was to explore shared experiences, collective norms, and group dynamics (e.g., community members, teachers, pregnant women), as this method encouraged interaction and exchange among participants from the same community groups. Thanks to the reviewer this has been clarified within the manuscript (lines 154 to 157).

• Lines 156–163: Justify differences in the endemicity levels and standardize their characterization across the selected areas in the three countries. For instance: Indonesia had no sampling from non-endemic areas, unlike Rwanda and Peru. Peru only sampled highly endemic areas, while Indonesia included low-, medium-, and high-endemicity areas. For Rwanda the endemic areas are mentioned without specifying the endemicity level. Thanks for this suggestion. We have now included a table to categorize better study locations across countries (Table 1). Not all locations (rural, urban, malaria endemic, and non-endemic) in the three countries had participants available for SSI/FGD in all types of areas. In addition, limited resources restricted the sampling to the ones that were more relevant for country context. We have clarified this in the revised manuscript (Lines 156–163).

• Lines 156–163: The selected areas in Indonesia and Rwanda lack description as urban or rural, while those in Peru are clearly characterized. I recommend standardizing this classification across all study sites for better comparability. Thanks for this suggestion. As stated above, we have now included in a table to categorize better study locations across countries (Table 1). (Lines 156–163).

• Lines 156–163: I suggest to present site characteristics (including FGD/SSI counts and sample sizes) in a table for a better overview of the sampling. Thanks to this suggestion, we have included a Table within the Methods section to explain the sampling procedures.

• Line 162: Fix spacing in "and Tumbes". Addressed

• Line 164: What is the reason for discussing Peru-specific findings separately and excluding them from this manuscript? Our partners in Peru were highly motivated to publish a manuscript focused specifically on the findings from the Peruvian context (Nussbaum, L., Ortega, E., Ríos López, E.J. et al. Voices from the Amazon: exploring implementor and user perceptions of non-invasive malaria diagnostics in Peru. Malar J 24, 32 (2025). https://doi.org/10.1186/s12936-025-05273-1). While those data are also included in the current submission, the focus of this manuscript is broader, presenting a comparative analysis across three countries. In contrast, the Peru-focused publication offers an in-depth examination of local findings unique to that setting. Partners in Indonesia are already preparing another separate country-focused manuscript to explore in depth country context perceptions.

• Line 180: Fix spacing in "aged 18 years". Addressed

• Line 252: I recommend sharing SSI and FGD guides as supporting information for more transparency of the applied methods. Thanks for this suggestion. We have included the guides as supporting information.

• Lines 259–261: Was the information in Figure S1 the only explanation given to participants about the devices? Was the figure translated into local languages? Were the devices limitations or additional details provided? The devices were presented following the SSI and FGD guides. We have now included that information

---

## [Editor Report · Decision Letter 1]

1 Oct 2025

*“It doesn’t require blood!”*

Dear Dr. Calarco,

We look forward to receiving your revised manuscript.

Kind regards,

Susan Horton

Academic Editor

PLOS ONE

Additional Editor Comments :

The line numbers below correspond to the “track changes” version.

Line 80: I don’t believe it is essential to describe UNITAID in detail, can omit the explanatory sentence.

Line 82: where is the reference to the technical landscape assessment by FIND?

Line 159: suggest small rewrite “and professionals working at borders who, because of human mobility, were familiar with malaria policies”

Line 210: Why do border professionals appear both in row 2 and row 3 of Table 2. Should they not appear in only one row?

Line 238: Suggest “These are…” not “Are”

Line 307: suggest amend sentence to say “there were 7 FGDs in Peru, 5 in Rwanda and 4 in Indonesia”. The total numbers of participants appear in the table and do not need to be repeated in the text.

Line 416: How were the interviewees made aware of the characteristics of the new tests, in particular the fact that they could not differentiate different species of malaria parasites? Were they shown Figures 1 and 2, or how was this information transmitted?

Line 642-651: I find this paragraph unnecessarily detailed, especially since there is no similar paragraph for Rwanda and Peru. Perhaps rephrase: “In Indonesia, participants highlighted the importance of Puskesmas (primary health centers), community health centers, and community health cadres (volunteers)”. This structure is very common in many low- and middle-income countries.

---

## [Author Response · Author response to Decision Letter 2]

15 Oct 2025

Additional Editor Comments :

The line numbers below correspond to the “track changes” version.

Line 80: I don’t believe it is essential to describe UNITAID in detail, can omit the explanatory sentence. Thanks, we addressed this by removing the sentence.

Line 82: where is the reference to the technical landscape assessment by FIND? We are currently updating the document and plan to submit it to a peer-reviewed journal. At this stage, however, the document remains internal and unpublished. For this reason, when referring to the UNITAID landscape we use the term “published”, while for the FIND landscape, we use “conducted by FIND”. We hope this clarification is clear.

Line 159: suggest small rewrite “and professionals working at borders who, because of human mobility, were familiar with malaria policies”. Thanks for the suggestion, we reviewed accordingly.

Line 210: Why do border professionals appear both in row 2 and row 3 of Table 2. Should they not appear in only one row? We apologize to the reviewer, but we don’t see “border professionals” appearing twice; maybe is a formatting issue from the track change version.

Line 238: Suggest “These are…” not “Are”. This point has been addressed at row 226 of the revised tracked-changes document.

Line 307: suggest amend sentence to say “there were 7 FGDs in Peru, 5 in Rwanda and 4 in Indonesia”. The total numbers of participants appear in the table and do not need to be repeated in the text. This point has been addressed at row 295 of the revised tracked-changes document.

Line 416: How were the interviewees made aware of the characteristics of the new tests, in particular the fact that they could not differentiate different species of malaria parasites? Were they shown Figures 1 and 2, or how was this information transmitted? Figure S1, included in the supplementary information, was used to guide the discussion and provide an illustration of how the devices might look. Additional details, such as their capacity to differentiate between malaria species, were conveyed orally during the interviews within participants questions, answers and discussion. In S3 Guide for the FGDs you can see how the question regarding the parasite species was asked “What are the key characteristics that each instrument shall have? (List key characteristics quoted by each participant and check if the ability to differentiate parasite species emerged. If not, ask the question: Do you think it would be relevant if the instrument can differentiate between parasite species (Vivax, Falciparum, Ovale, Malaria, knowlesi)?” for participants' exploration.

Line 642-651: I find this paragraph unnecessarily detailed, especially since there is no similar paragraph for Rwanda and Peru. Perhaps rephrase: “In Indonesia, participants highlighted the importance of Puskesmas (primary health centers), community health centers, and community health cadres (volunteers)”. This structure is very common in many low- and middle-income countries. Thanks for this suggestion, we have reviewed the text to be less specific. Now it reads: “In Indonesia, participants highlighted the importance of primary health centers, that offer a wide range of basic healthcare services, community health centers that serve as the first point of contact for most, and health cadres with authorized volunteers to administer malaria medication, especially in remote areas.”

---

## [Editor Report · Decision Letter 2]

20 Oct 2025

*“It doesn’t require blood!”* : Perceptions around non-invasive malaria testing tools in Indonesia, Peru, and Rwanda

PONE-D-24-60442R2

Dear Dr. Calarco,

We’re pleased to inform you that your manuscript has been judged scientifically suitable for publication and will be formally accepted for publication once it meets all outstanding technical requirements.

Kind regards,

Susan Horton

Academic Editor

PLOS ONE
---

## [Editor Report · Acceptance letter]

PONE-D-24-60442R2

PLOS One

Dear Dr. Calarco,

I'm pleased to inform you that your manuscript has been deemed suitable for publication in PLOS One. Congratulations! Your manuscript is now being handed over to our production team.

Kind regards,

on behalf of

Dr. Susan Horton

Academic Editor

PLOS One